# Large-scale predictions of alternative protein conformations by AlphaFold2-based sequence association

Myeongsang Lee [1,3], Joseph W. Schafer[1,3], Jeshuwin Prabakaran[1], Devlina Chakravarty[1], Madeleine F. Clore [1] & Lauren L. Porter [1,2] ✉

The many successes of AlphaFold2 (AF2) have inspired methods to predict multiple protein conformations, many of which have biological significance. These methods often assume that AF2 relies on evolutionary couplings to predict alternative protein conformations, but they perform poorly on fold-switching proteins, which remodel their secondary structures and modulate their functions in response to cellular stimuli. Here we present a method designed to leverage AF2's learning of protein structure more than evolutionary couplings. This method–called CF-random–outperforms other methods for predicting alternative conformations of not only fold switchers but also dozens of other proteins that undergo rigid body motions and local conformational rearrangements. It also enables predictions of fold-switched assemblies unpredicted by AlphaFold3. Several lines of evidence suggest that CF-random sometimes works by sequence association: relating patterns from homologous sequences to a learned structural landscape. Through a blind search of thousands of *Escherichia coli* proteins, CF-random suggests that up to 5% switch folds.

Alternative protein conformations can play critical roles in protein function and regulation[1–3]. These alternative conformations can be accessed by rigid body reorientations, local fluctuations, or remodeling secondary and/or tertiary structure (fold switching)[4]. Physics-based methods, such as molecular dynamics (MD) simulations, have successfully modeled alternative conformations[5–8], but they require too much computational power to predict conformational changes on a large scale. Furthermore, some conformational changes, such as fold switching, occur on a timescale of seconds[9–11], prohibitively long for MD to reasonably access alternative conformations if they were not known previously. Recently, artificial intelligence (AI)-based protein structure predictors–particularly AlphaFold2 (AF2)[12]–have offered another way to predict large numbers of alternative protein conformations[13–17]: modifying or subsampling the inputted multiple sequence alignment (MSA), from which AF2 may infer evolutionarily

coupled residue pairs used to predict structure[15]. These MSA modifications are hypothesized to diminish dominant residue-residue couplings[13,18] while sometimes enhancing the couplings of conformational alternatives[16,17]. However, a recent study found that these methods usually fail on the alternative conformations of 92 experimentally characterized fold-switching proteins likely in AF2's training set with individual false negative failure rates from 80-93%[19].

Here, we present CF-random[20], an alternative strategy to predict alternative protein conformations. This strategy leverages ColabFold[21] (CF)–an efficient-yet-accurate implementation of AF2–to predict alternative conformations by randomly subsampling input MSAs at depths too shallow for robust coevolutionary inference. While CF-random was shown to perform well on eight fold-switching proteins[20], its generalizability was not tested, its inner-workings were not explained, and its implementation was not automated. Here, we

[1]National Center for Biotechnology Information, National Library of Medicine, National Institutes of Health, Bethesda, MD 20894, USA. [2]Biochemistry and Biophysics Center, National Heart, Lung, and Blood Institute, National Institutes of Health, Bethesda, MD 20892, USA. [3]These authors contributed equally: Myeongsang Lee, Joseph W. Schafer. ✉e-mail: porterll@nih.gov

# CF-random

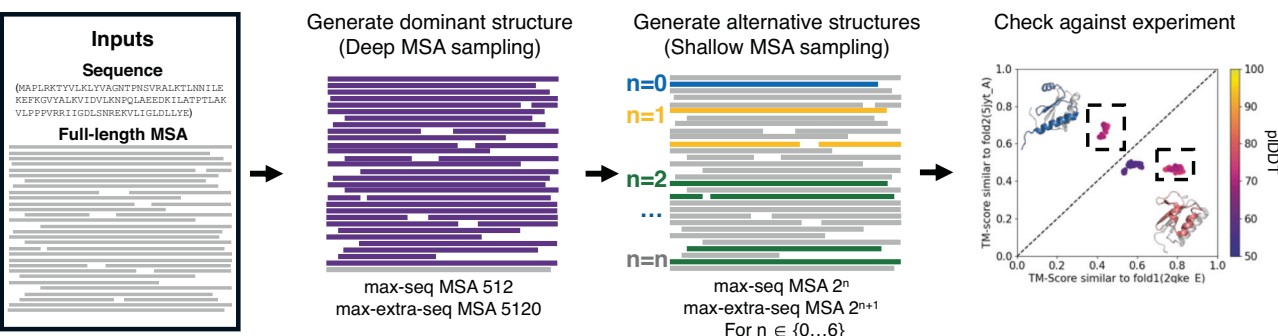

**Fig. 1 | Overview of CF-random.** CF-random (ColabFold-based random MSA sampling) generates dominant and alternative protein structures by combining ColabFold predictions generated from a deep multiple sequence alignment (MSA) and shallow random MSAs, respectively. To run default mode, a full-length MSA is required. ColabFold samples the MSA at default depth (512:5120) to produce the dominant conformation and then randomly subsamples at shallower depths to predict putative alternative conformations. It was benchmarked against known fold-switching proteins by calculating the TM-scores of all predicted structures against two reference structures. Success was considered correct predictions of both conformations from a single target sequence. Source data are provided as a Source Data file.

address all three of these issues by (1) testing CF-random on 92 fold-switching proteins and 37 other proteins that undergo local conformational fluctuations and rigid body motions, (2) showing that CF-random sometimes works by sequence association: relating patterns from homologous sequences to a learned structural landscape, and (3) providing an automatic implementation. We find that CF-random outperforms all methods reported previously. Compared to the 7−20% success rates of other individual methods for fold-switching proteins, CF-random achieved a 35% success rate while generating 6x fewer structures overall. Further, CF-random captured rigid body reorientations and local conformational fluctuations of the 37 other proteins with a 95% success rate and considerably less sampling than other methods. Encouraged by its success, we develop CF-random to perform blind searches for alternative conformations and run it on >2000 proteins from *Escherichia coli*. These predictions suggest that up to 5% of *E. coli* proteins switch folds. We present CF-random as a tool for community use, specifying its strengths and limitations.

## Results

### CF-random outperforms other AF-based predictors of fold switching

CF-random is a ColabFold-based pipeline that generates putative conformational ensembles by combining predictions from deep and shallow MSA sampling (Fig. 1, **Methods**). AF2-based methods, such as ColabFold, are known to generate one dominant conformation of fold-switching proteins from deep MSAs[22]. Thus, the challenge is to sample the alternative conformation. CF-random aims to overcome this challenge by sampling very shallow random input MSAs with as few as 3 sequences. Shallow sampling directs the AF2 network to predict structures from sparse sequence information insufficient for robust coevolutionary inference, setting it apart from previously proposed methods that used a minimum of 24 sequences[14,23]. Because some predictions are not well folded at shallow MSA sampling depths, CF-random also explores deeper depths. Typically, shallow sampling occurs at seven depths including between 3 and 192 sequences. Template modeling (TM)-scores[24] of the fold-switching regions of each prediction are compared to their experimentally determined structures, since this has been shown to discriminate between fold switchers better than overall TM-score[19], though the overall score is considered also. We report sampling depth with the following notation: x:y, where x is the value assigned to ColabFold's −max-seq

argument (the number of sequences randomly selected as cluster centers) and y is the value assigned to −max-extra-seq, the number of extra sequences randomly sampled from the clusters is defined by −max-seq. The total number of sequences inputted into ColabFold at each recycling step is x + y.

We tested CF-random on 92 fold-switching proteins likely in AF2's training set and found that it predicts both the dominant and alternative conformations of 32 fold switchers successfully. Further, CF-random sampled 89% fewer structures than other AF2-based methods (Fig. 2a), which predicted 25 fold switchers altogether. To explore what AF2 has learned about alternative conformations of fold switchers, all predictions were performed without templates. Very shallow sequence sampling was a key to CF-random's success: 23 conformations (72%) were successfully predicted at sampling depths of 4:8 sequences or below (Fig. 2b). To our knowledge, random sequence sampling at such shallow depths has not been tested systematically; previous work suggested a minimum sampling depth of 8:16[14,23].

CF-random successfully predicts both global and local fold-switching events, some of which had not been predicted successfully by other methods (Fig. 2c). For instance, CF-random successfully predicts both conformations of human XCL1, which have distinct hydrogen bonding networks and hydrophobic cores[25]. Though both predictions correspond well with experimentally determined structures, CF-random mispredicts that a disordered C-terminal region of dimeric XCL1 folds into a helix, consistent with other reports that AF2 sometimes incorrectly predicts disordered regions as helices[26]. CF-random also predicts two conformations of TRAP1-N. This N-terminal domain of a human mitochondrial heat shock protein assumes different conformations in its apo, GTP-, and GDP-bound forms[27]. The alternative autoinhibitory apo form has a conserved glutamine that binds to the ATP-binding lid region and is proposed to be an on-pathway conformation important for TRAP1N's function as a protein folding chaperone. The dominant ATP-open form has not been observed to promote protein folding. Finally, RepE, a DNA replication initiator protein from *E. coli*, has two folds with distinct functions. As a monomer, it functions as a replication initiator whereas the dimeric form functions as a repressor. Its dimeric form is dominant under physiological conditions; the chaperone DnaK mediates its conversion to a monomer[28]. CF-random predicts its monomeric form from full MSAs and its dimeric form at a very shallow sampling depth (2:4, Fig. 2c).

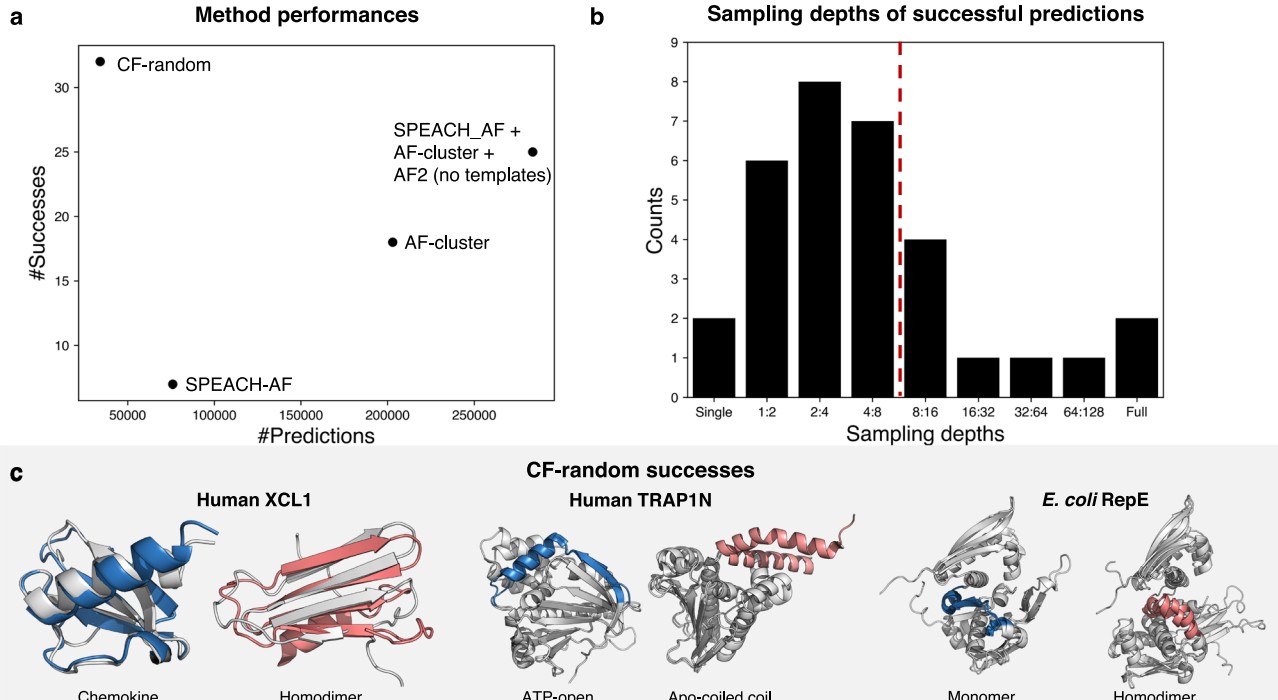

**Fig. 2 | CF-random outperforms other methods for predicting fold switching. a** CF-random performs more successfully and efficiently than SPEACH-AF, AF-cluster, and AF2 with no templates. **b** Most alternative conformations can be predicted successfully using very shallow sampling depths (those to the left of the dotted line comprise 72% of all predictions). In two cases, sampling at 1:2 produced homogenous well-folded structures similar to the dominant conformation, in which case single sequences were also tested (single). **c** Successful predictions of three fold-switching proteins not predicted by the other methods: human XCL1, human TRAP1N, and *Escherichia coli* (*E.coli*) RepE. Gray represents the reference PDB structures and single-folding regions; blue and pink indicate the fold-switching regions of dominant and alternative structures, respectively. Source data are provided as a Source Data file.

## Combining AF2-multimer with CF-random sampling improves some predictions

Of the 32 successful predictions of fold switchers, six were improved from additional molecular context supplied by the AF2-multimer model. For instance, CF-random predicted a monomer of FraC–a pore-forming toxin from *Actinia fragacia*–to have its N-terminal helix detached from the rest of the protein, similar to the pore form[29] (Supplementary Fig. 1a). Inputting its sequence into AF2-multimer and sampling at the shallowest depth found to produce folded predictions yielded a structure highly consistent with experiment (1.4 Å, Fig. 3a). Interestingly, neither sampling deeper MSAs nor using the AlphaFold3[30] (AF3) server produced this conformation (Fig. 3a). Furthermore, though AF2-based predictors often struggle to predict amyloid structures[31,32], CF-random produced an amyloid-like structure of the human Aβ42 peptide (Supplementary Fig. 1b). Running the multimer model on its sequence at the shallowest depth possible (1:2) yielded a fibril-like conformation consistent with experiment (Fig. 3b). While the morphology of the overall fibril was not completely correct–the experimentally determined structure involved three chains while the prediction involved two in a similar configuration–the prediction suggests that an amyloid-like configuration is possible. Deep-MSA sampling with the AF2 multimer model did not produce amyloid-like fibrils consistent with those in the PDB by a Foldseek[33] search, and AF3 predicted a different fibril morphology also loosely consistent with experiment (Fig. 3b). Additionally, CF-random predicted a partially folded conformation of the cell cycle regulatory protein Cks1 from *S. cerevisae* consistent with its functionally relevant domain-swapped dimer[34] (Supplementary Fig. 1c). Running the AF2 multimer model on both deep and shallow MSAs produced an experimentally consistent dimer orientation, while AF3's was inconsistent (Fig. 3c). The multimer model also enhanced promising predictions of the domain-swapped conformation of human CrkL-SH3C domain (not predicted by AF3) and domain-swapped *Escherichia coli* rhomboid protease and *E. coli* FimF (both predicted by AF3). After running CF-random with multimer weights on all 92 fold-switching proteins, only one additional alternative conformation was found: the dimeric form of bone morphogenic protein inhibitor DAN from *Mus musculus* (PDB ID: 4jph). It was not included among the successes since it was not identified when CF-random was run on a single protein chain.

## CF-random predicts rigid body motions and local conformational changes more efficiently than other AF-based methods

CF-random's performance on alternative conformations of fold switchers raises the question of how well it predicts other conformational changes. To address this question, we tested it on two other datasets used to assess how well AF2-based methods predict alternative conformations. The first was a dataset of 14 proteins, 2 soluble proteins and 12 membrane transporters used to benchmark two previous methods[14,16]. Like CF-random, one of these two methods (SPEACH-AF) did not use templates to make predictions. SPEACH-AF works by making in silico alanine mutations to columns of multiple sequence alignments (MSAs) corresponding to groups of residues in direct contact. The idea is that these alanine mutations mask dominant evolutionary couplings used for structural inference, allowing AF to detect alternative couplings invisible in the full unmasked MSA[16]. The other dataset contained 23 proteins with well-defined open and closed conformations (OC23), such as periplasmic binding proteins, used to benchmark another method called AFSample2[13]. Like SPEACH-AF, AFSample2 also masks columns of MSAs but randomly rather than targeting residues in direct contact. It also enables dropout, which set some of AF2's weights to 0 during inference, making it easier for the network to sample uncertainties and generate more diverse structures.

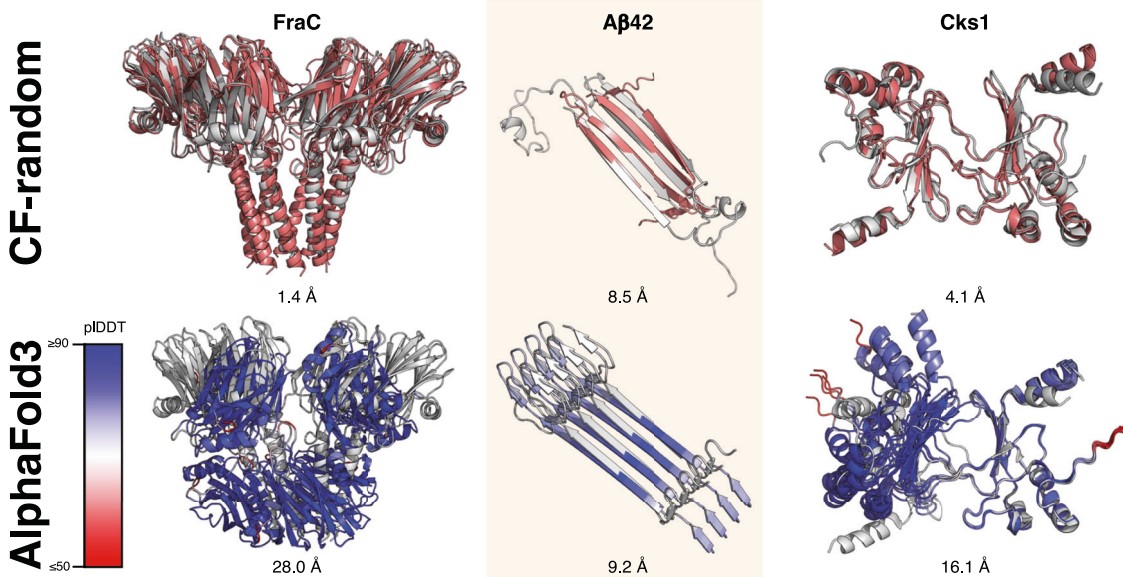

**Fig. 3 | Running CF-random with AF2 multimer weights produced experimentally consistent protein assemblies.** By contrast, AlphaFold3 (AF3) predicted some assemblies incorrectly but with high confidence. For CF-random, predictions are pink, denoting alternative conformations, while AF3 predictions are colored by plDDT. Experimentally determined predictions are gray with PDB IDs: 4TSY (FraC), 6RHY (Aβ42, above), 2MVX (Aβ42, below), 1QB3 (Cks1). Experimentally determined structures are gray. Source data are provided as a Source Data file.

CF-random sampled alternative conformations more efficiently than both methods while producing accurate predictions. For instance, it successfully captured local conformational fluctuations in the "LID" and "AMP-binding" domains of *Escherichia coli* Adenylate Kinase (AK) with high accuracy (TM-scores of 0.98 and 0.9 for the dominant and alternative forms, respectively, Fig. 4a,b). Overall, CF-random captured both Fold1 and Fold2 of all SPEACH-AF targets with TM-scores ≥ 0.81 with considerably less sampling than SPEACH-AF: 200 structures/CF-random ensemble compared to 300-811 structures/SPEACH-AF ensemble (Fig. 4c,d, Supplementary Table 2). Similarly, CF-random captured the "cap-open" and "cap-closed" states of β-phosphoglucomutase (βPGM) with TM-scores of 0.97/0.99 for the dominant/alternative conformations (Fig. 4e,f). Overall, CF-random captured both Fold1 and Fold2 of all OC23 targets with TM-scores ≥ 0.7, except for Q9ERE7 (Fig. 4g); AFSample2 also failed to predict its conformations with high accuracy (Supplementary Table 3). However, CF-random successfully captured three other alternative conformations that AFSample2 failed to capture (Supplementary Table 3). Further, CF-random sampled 5x fewer models to predict alternative conformations compared to AFSample2 in 16/23 cases (Fig. 4h). In 6/7 remaining cases, sampling an additional 600 structures yielded high accuracy predictions of both conformations, 20% less sampling than AFSample2 (Fig. 3h).

### Sequence association drives some predictions of alternative conformations

How does CF-random predict alternative conformations? Since AF2 predictions are often based on evolutionary couplings inferred from multiple sequence alignments (MSAs), it has been proposed that supplying varied MSAs can provide evolutionary restraints unique to alternative conformations[15,17]. Recent work indicates that this is not the case for rigid body motions and local conformational fluctuations, however[18]. Instead, conformations consistent with the same set of evolutionary restraints are sampled stochastically. Fold switching differs from rigid body motions and local conformational fluctuations because it involves remodeling of secondary structure, sometimes leading to alternative folds inconsistent with dominant evolutionary couplings[20]. Since AF2-based predictions of dominant fold-switch conformations are typically consistent with coevolutionary restraints from deep MSAs, we investigated whether successful AF2 predictions of fold switchers are driven by coevolutionary restraints unique to their alternative conformations.

Coevolutionary analysis of the MSAs that successfully produced each alternative fold-switched conformation revealed few unique alternative evolutionary couplings, indicating that predictions of alternative conformations of fold switchers are not generally driven by coevolutionary inference (Fig. 5a). In fact, these MSAs provide more information unique to dominant unpredicted conformations (mean percentage of total couplings, 11%) than alternative predicted conformations (mean percentage of total couplings, 8%). Thus, many of these predictions seem to be driven by AF2's learning of protein structure rather than couplings unique to alternative conformations. Since previous work suggests that (1) AF2 struggles to predict the large conformational changes that many fold switchers undergo[18] and (2) AF2 has memorized some alternative conformations of fold switchers[19,20], we hypothesized that AF2 may generate alternative conformations of fold switchers through sequence association: relating features of homologous sequences to a learned structural landscape. This differs from coevolutionary inference, which can be used to infer protein structure without prior knowledge.

We used a recently characterized fold switcher, Sa1 V90T (Sa1 hereafter), to test whether AF2 may produce some alternative conformations through sequence association. Sa1 is a temperature-sensitive fold switcher that assumes a three-α-helix bundle fold at low temperature but switches to an α/β plait as temperature increases[13]. Some methods for predicting multiple conformations struggle to predict both folds of Sa1[19]. Similarly, with default settings, CF-random predicts the α/β plait fold but not the helical bundle. Investigating, we noticed that although the initial iteration of sequence search for Sa1 yielded sequences unique to both folds, subsequent sequence-search iterations left its final MSA without three-α-helix bundle sequences (Fig. 5b).

We then searched the sequence of Sa1 against the PDB and found it to match 3 three-α-helix folds with sequence identities ranging from 42−70%; these sequences revealed that CF predicts the three-α-helix fold through sequence association. In further detail, we ran CF three

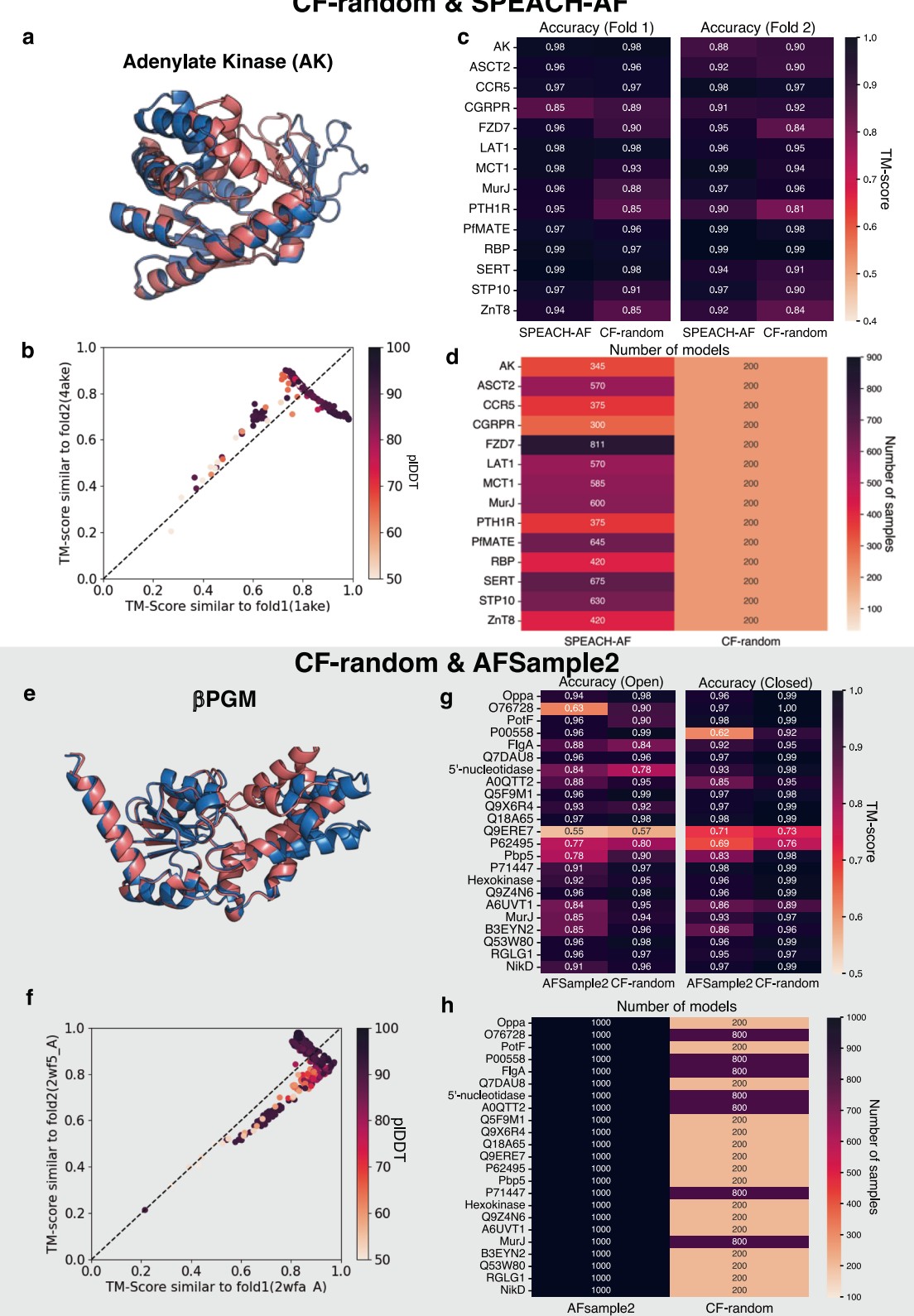

**Fig. 4 | CF-random efficiently and accurately predicts rigid body reorientations and local conformational changes. a, e** The open- and closed-conformations of Adenylate Kinase (AK) and β phosphoglucomutase (βPGM) were correctly predicted. Dominant/alternative conformations are blue/pink. **b, f** CF-random produces ensembles of AK and βPGM accurately and with high confidence, though some inaccurate lower confidence structures are produced also. **c, g** The highest TM-scores that CF-random produced are reported in these tables and compared with SPEACH-AF and AFSample2, respectively. **d, h** CF-random sampled substantially fewer structures than either SPEACH-AF or AFSample2. Source data are provided as a Source Data file.

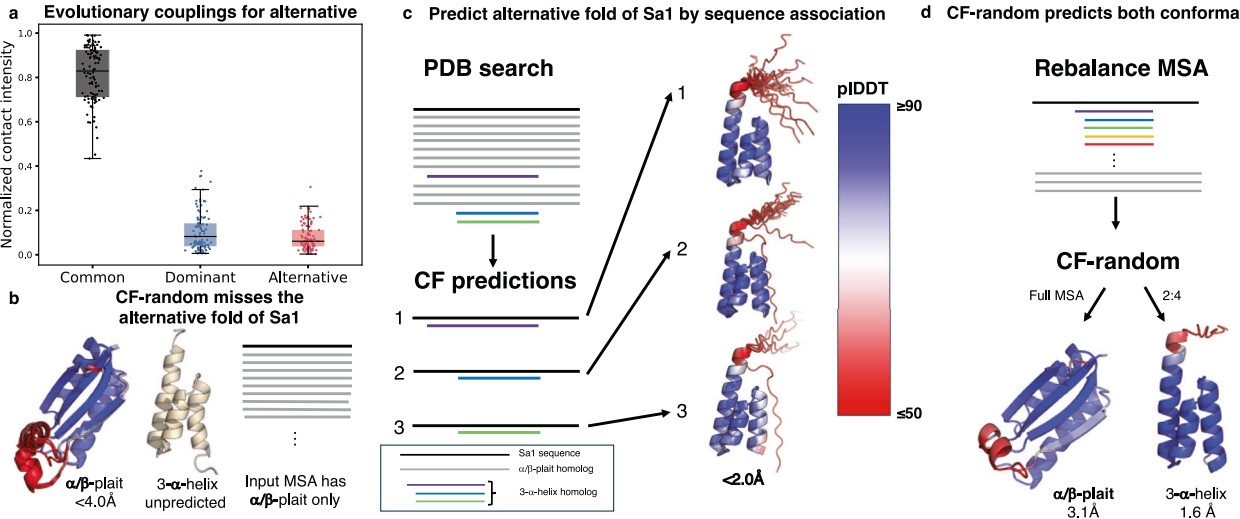

**Fig. 5 | CF-random predictions of alternative conformations are sometimes driven by sequence association. a** Evolutionary couplings unique to dominant unpredicted structures are generally stronger than evolutionary couplings unique to predicted alternative structures. Each box-and-whisker plot contains 104 data points, with boxes representing interquartile ranges and the middle line representing the median. Whiskers extend to 1.5 times the interquartile range. **b** CF-random failed to predict the alternative conformation (3-α-helix bundle fold) of Sa1 using the multiple sequence alignment (MSA) with α/β-plait homologs only. **c** To test for sequence association, three different MSAs were generated containing the Sa1 sequence and a single 3-α-helix homolog over the protein data bank (PDB)

search. In all three cases ColabFold (CF) predicted the helical bundle sequence with high confidence, indicating that they are predicted by sequence association. **d** When the MSA was rebalanced with 3-α-helix (orange and red sequences also represent 3-α homologs) and α/β-plait, CF-random successfully predicts both Sa1 structures. It predicted the α/β-plait from the full MSA and the 3-α-helix bundle with --max-seq = 2, --max-extra-seq = 4. Color bar representing AF2's confidence metric, per-residue local distance difference test (plDDT) scores, applies to all structures with corresponding colors. Source data are provided as a Source Data file.

times, inputting only one of the three PDB sequences in each run. In all three cases, CF confidently predicted three-α-helix folds from the two-sequence MSAs (Sa1 sequence+homolog, Fig. 5c). Since these two-sequence MSAs are too shallow for robust coevolutionary inference (Supplementary Fig. 2), it seems the best explanation for these predictive successes is sequence association. This approach is much like homology modeling but more powerful because only a single "template" sequence–rather than an input structure–is needed to produce the alternative conformation. Underscoring the explanatory power of this approach, it produces the helical bundle conformation more frequently (40-100% of the time from 20 models, depending on input sequence) than in previous work sampling 1000 structures with dropout and MSA masking (<2% of the time)[13]. Some of these three-helix-bundle predictions also contained a fourth C-terminal helix experimentally observed to be disordered, again demonstrating that sometimes AF2 sometimes mispredicts disordered regions as helical.

Leveraging these observations, we developed a strategy that enabled CF-random to predict both folds of Sa1 (Fig. 5d). First, Sa1's MSA was rebalanced to contain hundreds of 3-α-helix bundle homologs but only 3 sequences homologous to the α/β-plait. Interestingly, sampling the modified MSA at full depth produced α/β-plait models with high confidence and RMSDs closer to the experimentally determined structure than the deep MSA with >1000 α/β-plait sequences. Since AF2 predicts the α/β-plait fold of Sa1 from a single sequence with high confidence (Supplementary Fig. 3), sequence association seems to be the best explanation for this prediction. Likewise, sampling at shallow depths such as 2:4 yields high-confidence and accurate predictions of the three-α-helix fold (Fig. 5d). Given the very shallow sampling depth and the observations in Fig. 5c, sequence association seems to be a sensible explanation for this prediction as well.

**Blind predictions with CF-random**
CF-random was further developed to predict new fold switchers without prior knowledge. This approach extends the algorithm by

automatically selecting putative alternative conformations among the predicted models; the remainder of the algorithm is unchanged. In both previous work and this study, we noticed that correctly predicted alternative conformations of fold switchers are not always assigned confident scores by AF2[19], as measured by plDDT (per-residue local Distance Difference Test). For instance, the alternative fold of XCL1 is predicted with low confidence (plDDT <70) in most cases (Supplementary Fig. 4); low plDDT scores of alternative conformations have been observed in other work as well[18,35]. To circumvent this problem, we developed an approach to cluster predictions by structural similarity (Fig. 6). Although structural clustering can be straightforward for proteins whose alternative conformations differ substantially from dominant, it is more challenging when structural differences are localized to relatively small protein regions. To overcome this challenge, a Foldseek[33] database of all structures predicted for a given target is constructed. This database is used to calculate an all against all similarity matrix for the structures using the Foldseek structural bitscore, which enables both subtle and large conformational differences between predicted structures to be distinguished (Methods). The similarity matrix is then reduced using Principal Component Analysis (PCA) followed by two different density-based scanning algorithms, finally yielding a subset of the full CF-random blind mode ensemble that represents the conformational variance of the predicted structures. Blind mode successfully detected both conformations of fold switchers in 81% of cases identified in default mode (26/32), corresponding to a 28% success rate among all 92 fold switchers. An example of a difficult-to-identify local conformational change is the active-site loop of inositol monophosphatase (IMPase) from *Thermatoga maratima* shown in Fig. 6. Well-folded predictions of both conformations are present in the yellow and green clusters. Structures from purple and blue clusters are not as well folded (Supplementary Fig. 5). Thus, user discretion is required to identify the most plausible alternative conformations among those suggested by blind mode.

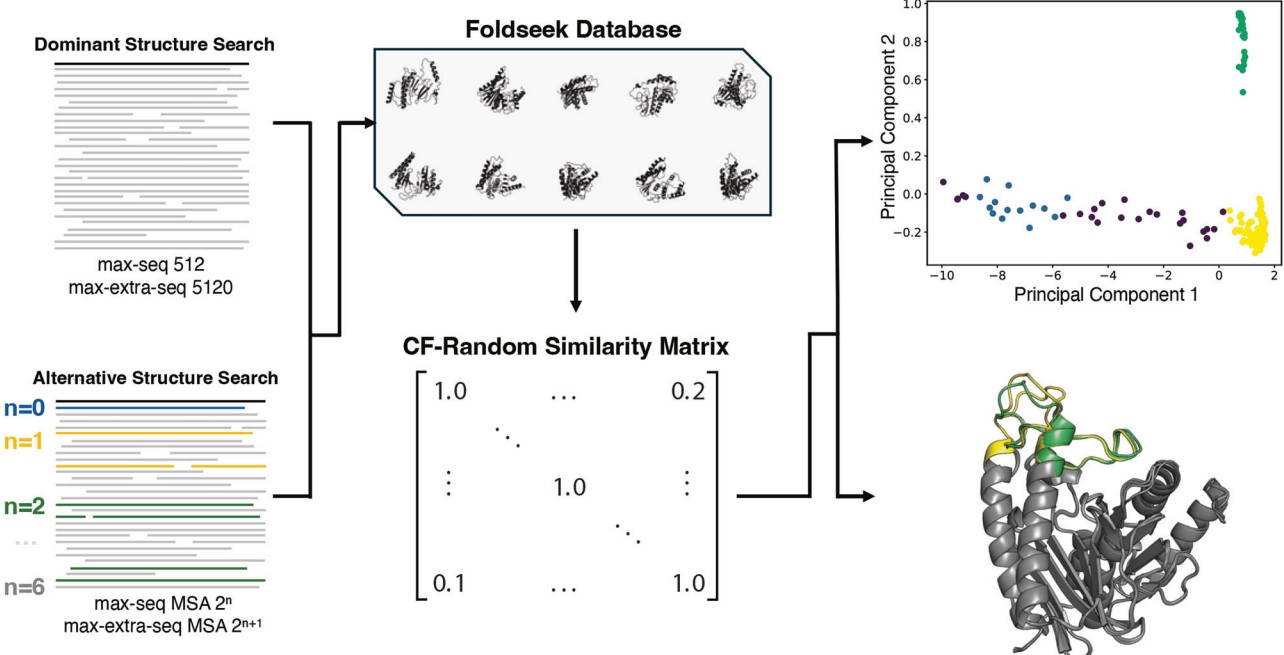

**Fig. 6 | Flow diagram of CF-random's blind mode.** A Foldseek database is created from CF-random generated structures (collectively called the ensemble). This database is used to evaluate each structure's similarity to the ensemble. Contrasting default mode in Fig. 1, two reference structures are not required, enabling a blind search. Principal component analysis followed by two different clustering algorithms produce the final subset of CF-random conformations that represent the structural variance in the ensemble. From left to right: CF-random blind mode produces 200 predicted structures; all structures are queried against the generated database to create the similarity matrix. Predicted structures from T. maratima IMPase (PDB ID: 2P3V) are depicted within the similarity matrix representation. Principal component analysis and HDBSCAN then produce a reduced space with similar structures clustered near each other (scatter plot top right), and K-medoids selects representative structures from the HDBSCAN clusters (2P3V predicted structures for the green and yellow clusters are shown bottom left, purple and blue structures in Supplementary Fig. 5). Source data are provided as a Source Data file.

## Blind predictions of fold-switching E. coli proteins

Finally, we tested CF-random's blind mode on *E. coli* proteins–including some from bacteriophage–with 300 residues or fewer (Fig. 7a). This length limit makes them suitable for future experimental testing by nuclear magnetic resonance (NMR) spectroscopy, a method that has successfully identified fold-switching proteins previously[36]. To ensure maximum plausibility, predictions reported here were also subject to the following quality checks.

(1). Coevolutionary evidence for both conformations. Both dominant and alternative protein conformations often have clear coevolutionary signatures[37]. As shown previously, AF2-based predictions, including those by CF-random, do not necessarily result from coevolutionary inference[18–20]. Predictions of alternative conformations without evolutionary basis can be incorrect[20,37]. Therefore, we searched for coevolutionary evidence of both conformations using Alternative Contact Enhancement (ACE), a method designed to identify evolutionary couplings unique to multiple protein conformations[37]. If such evidence was found, a prediction was considered plausible if condition (2) was met. We also note that at shallow depths, CF-random can predict spurious helical conformations. Cross-checking against coevolutionary information often helps to eliminate them. In cases where insufficient evolutionary couplings were present, we used our best judgement to assess the predictions.

(2). Ruling out interchain contact misassignment. Previous work has shown that AF3 can misassign intrachain contacts as interchain[19]. Here, we observe the opposite problem: sometimes AF2 assigns interchain contacts as intrachain (Supplementary Fig. 6), producing questionable alternative conformations. To circumvent this problem, we performed Foldseek searches on all hits produced by blind mode and examined their oligomeric states. If the target had a close homolog (≥70% sequence identity) with an oligomeric state whose interchain contacts overlapped with contacts unique to the predicted alternative conformation, the prediction was discarded under the assumption that it forms an oligomer and its interchain contacts were misassigned as intrachain, producing a spurious alternative conformation. In cases where a target did not have a close homolog with solved structure, but both of its conformations had coevolutionary support, it was reported. These cases may also conflate intrachain contacts with interchain but lacked evidence to support exclusion.

(3). Confidence ranking. Most confident (Tier 1) predictions satisfied criteria (1) and (2) while also being supported by experiment or having a strong biological basis for the conformational change. Tier 2 predictions satisfied criteria (1) and (2) without further experimental support.

With this approach, we ran CF-random on 2126 *E. coli* proteins, of which 52 were predicted to switch folds (Supplementary Table 4). Supporting the plausibility of these predictions, the estimated relative Rosetta energy scores[38,39] between each pair of putative fold switchers were in the same range as fold switchers determined by experiment (Supplementary Fig. 8). Among the functionally annotated putative switchers, transcription/translation regulators were the most common, followed by toxin-antitoxin proteins, phage proteins, and structural assembly proteins (Fig. 7b). Toxin-antitoxin proteins and proteins of unknown function have not been observed to switch previously, while previous work has established that proteins in the other functional classes sometimes switch, especially transcription factors, which are enriched among fold switchers. Furthermore, the blind search identified some experimentally confirmed fold switchers, including RfaH and MinE, along with homologs of fimbrial proteins known to switch folds.

Three putative fold switchers are presented as examples (Fig. 5c, Supplementary Fig. 7). The phage tail tube protein is part of a large

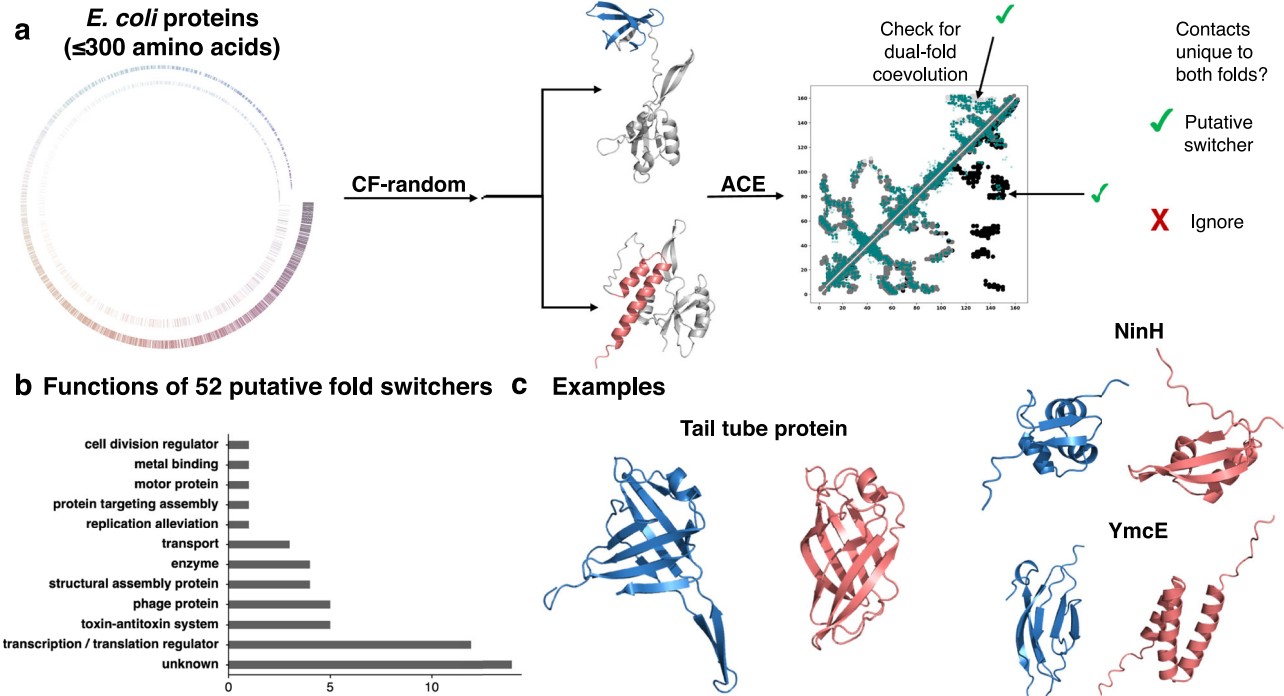

**Fig. 7 | Predicting E. coli proteins that may switch folds. a** We ran the sequences of 2126 E. coli and phage proteins through CF-random. In total 3237 sequences were attempted, but in 1111 cases, MSAs deep enough for coevolutionary inference could not be generated. Seashell-like image represents these 3237 proteins by length; the inner circle represents 1111 rejected candidates, and outer, the 2126 proteins that were then run through CF-random. If two or more distinct conformations were identified, such as in the case of the successfully identified fold-switching E. coli protein, RfaH, we tested for dual-fold coevolution using ACE. If coevolutionary evidence for both folds was found, the protein was considered a putative fold switcher. Light gray/black contacts on upper/lower diagonals are unique to CF-random predicted dominant/alternative conformations. Teal contacts are from coevolutionary inference. Medium gray contacts are unique to both folds.

**b** Putative fold-switching proteins are involved in diverse functions. **C** Examples of putative hits. The phage tail tube protein is part of a large assembly that penetrates its host cell envelope. The dominant conformation predicts an extended β-sheet, while in the alternative form, the sheets are incorporated into the larger β-sheet structure. Both conformations were predicted with plDDT scores > 70. NinH is transcription factor protein that may undergo an α-helix-to-β-sheet transition; both conformations were predicted with plDDT scores > 70. Finally, YmcE is a bacterial antitoxin predicted to assume two different folds. Its dominant form was predicted with plDDT > 80, while its alternative was predicted with plDDT > 66 excluding disordered ends. Throughout this figure, dominant folds are blue and alternatives are pink. Source data are provided as a Source Data file.

assembly that penetrates its host cell envelope. Its dominant form resembles the conformation of a homolog in its protein assembly, with protruding β-sheets that interact with its partners. Its alternative fold has its β-sheets tucked within the larger body of its structure and stabilized by hydrogen bonds. Though no similar PDB structures could be identified through a Foldseek search, this alternative conformation seems plausible given that similar behaviors in pore forming proteins have been observed[40]. Both conformations were predicted with plDDT scores ≥ 70. Second, NinH is a transcriptional regulator whose closest FoldSeek matches are Cro repressor proteins. The structures of Cro proteins have evolved over time, and switches have been engineered[41,42]. This, combined with the fact that several known fold-switching proteins are transcription factors[4], supports the prediction. Finally, the bacterial antitoxin YmcE may also switch folds. This is the most speculative of the three predictions since few YmcE homologs were used to construct its MSA, and its evolutionary couplings were noisy. However, the hit is reported because (1) CF-random predicts that several bacterial antitoxins switch folds, and (2) YmcE is predicted to assume two dramatically different conformations.

Based on these predictions, we estimate that up to 5% of E. coli proteins switch folds. Of the 2126 proteins in the set, 52 are predicted to switch folds, a 2.4% hit rate. However, CF-random predicted only 22/47 of known fold-switching proteins with ≤300 residues. If an equal number was missed here, 5.2% would switch folds. Furthermore, 45/92 know fold-switching proteins have > 300 residues. Again, a similar

ratio among E. coli proteins would suggest that up to 5% may switch folds.

## Discussion

Though AlphaFold2 often predicts single protein conformations with high accuracy, predicting alternative conformations remains a challenge[43]. Some recently developed methods[44,45] generate alternative conformations using new techniques such as the Distributional Graphoformer[46] and flow matching[46], though these have not been tested on many fold-switching proteins. A diffusion-based model called EigenFold was tested on fold switchers, but its performance was weak[47]. Several previous methods have relied on coevolutionary information to predict fold switching and other conformational changes, for both AlphaFold2[15] and ESMFold[48,49]. Here we show that leveraging AF2's learning of protein structure–including sequence association–outperforms previous methods for predicting fold-switching proteins. Furthermore, this approach is more efficient than previous methods for predicting rigid body motions and local conformational rearrangements. The generalizability of CF-random suggests that leveraging AF2's learning of protein structure offers more robust predictions of alternative conformations than coevolutionary inference of input MSAs, and sequence association may be a far-reaching explanation for how AF2 predicts alternative conformations of fold switchers.

CF-random has both advantages and limitations. First, it predicts some fold-switched conformations through AF2's learning of protein

structure rather than coevolutionary inference of input MSAs. To our knowledge, this is the first study showing that AF2 sometimes works by sequence association. Much like homology modeling, sequence association requires prior learning of protein structure. We suspect that this drives many CF-random predictions at very shallow sampling depths (4:8 and below). While shallow random sampling may foster CF-random's efficiency, it may also impede predictions of alternative conformations very different from those in the training set. CF-random may make new associations between input sequences and structures in its training set, however, as it did with Sa1. Accordingly, it may successfully reveal that certain sequences assumed to be single folders can assume yet-undiscovered alternative conformations in the training set. Previous work indicates that AF2-based methods can incorrectly predict single folders as fold switchers[20], however. This is a second notable limitation of CF-random: it may predict alternative conformations of proteins erroneously. For this reason, we suggest using alternative approaches–such as ACE[37], AF2-RAVE[50], or other MD-based approaches[51]–to cross-check predictions. Nevertheless, some predictions presented here may be incorrect, particularly those of proteins with unknown function. Thirdly, blind mode sometimes misses alternative conformations that CF-random predicts correctly.

Although CF-random outperforms other methods for predicting known fold switchers, it is a weak predictor (35% success rate), indicating that much work remains to be done in this challenging research area[52]. CF-random was run without templates and with much less sampling than other methods. More sampling and inclusion of templates may lead to a higher success rate, though we do not expect major improvements since most of the successful predictions reported here were also achieved by combining results from other AF2-based methods with a lot more sampling, and inclusion of templates yielded only a few more unique hits[19]. The CF-random approach may improve AF3's performance on fold switchers also, since it was also weak (8% success rate)[19].

Nevertheless, CF-random predicts alternative conformations in >50 *E. coli* proteins, suggesting that up to 5% switch folds. This substantial number supports previous work proposing that fold switching is a widespread natural phenomenon[4,53]. To demonstrate its potential significance, there are ~20,000 human genes. If 5% of them switched folds, that would be ~1000 human fold-switching proteins. Future work will examine the scope of fold switching in human and other proteomes.

To our knowledge, this is the first study to successfully predict many plausible 3D alternative conformations from thousands of protein sequences. While the focus of this study was on fold switching, CF-random predicted other conformational changes also, suggesting that it can be used to predict alternative conformations in general. Future experimental work will test whether these predictions of fold switchers are correct. We hope that CF-random will help to advance the new frontier of predicting conformational ensembles[52].

## Methods

### Datasets
Fold-switching proteins adopt at least two different conformations (e.g., active/inactive or apo/holo conformations). In the CF-random pipeline, the dominant conformation is typically defined as that which CF predicts most frequently from full MSA sampling, except in a few cases where memorization has been shown or is expected to overrule coevolutionary inference[19,35,54]. AF2/CF rarely predicts more than one conformation when sampling full MSAs[22]. By exclusion, the alternative conformation is defined as that which AF2 rarely–if ever–samples from full MSAs and therefore requires an alternative sampling strategy.

PDB IDs corresponding to both conformations of all proteins tested here (fold switchers and other conformational changes) are in Supplementary Tables 1/2/3; regions of proteins that switch folds are also reported, if applicable. These regions are well ordered, as

evidenced by having B-factors comparable to regions of solved protein structures that do not switch folds[19].

### Default mode prediction for predicting alternative conformations and fold-switching proteins
All CF-random predictions were run using ColabFold (CF) version 1.5.5 with monomer-ptm weights and all MSAs were constructed using the CF pipeline unless specified otherwise. Previous work showed that monomer weights can perform better than monomer-ptm[20], but *E. coli* genome proteins were run with monomer-ptm weights before this observation was considered; rerunning with monomer weights was too costly. When run with either set of weights, default mode requires an input MSA and two reference structures as input. It works as follows:

Step 1: **Sampling**. Multiple Sequence Alignments are sampled at default depth (512:5120) to generate structures using all 5 AF2 models and 5 random seeds. Then 7 shallow depths (1:2, 2:4, 4:8, 8:16, 16:32, 32:64, 64:128) are sampled with 5 random seeds, all 5 models. This leads to 200 predicted structures overall. Importantly, as shown by previous work, the default depth is expected to predict one conformation[22]; thus, finding the alternative is the challenge. Note that X:2*X refers to the --max-msa:--max-extra-msa ColabFold parameters.

Step 2: **Checking against experiment**. TM-scores of each prediction are calculated against two distinct experimentally characterized conformations. If predictions with TM-scores ≥ 0.6 for both conformations were generated, and the best TM-score relative to conformation 1 > its TM-score relative to conformation 2 and vice versa, the predictions were considered a success. In some cases, such as human lymphotactin, the TM-score threshold was lowered to account for variability in disordered regions.

### Fold-switching predictions
The total number of successes was the number of sequences from which both conformations could be generated; one conformation had a higher TM-score corresponding to Fold1 and the other to Fold2. Models whose accuracies increased by running the Multimer model were included, though their alternative conformations were required to be sampled as monomers also. The total number of structures generated by CF-random was (200)*154 proteins with distinct sequences (many fold switchers have a few mutations to stabilize one conformation or are truncated[4]) = 30,800 structures. Additionally, models of 100 proteins were generated from single sequences, 25 structures each = 2500 additional structures. In two cases where the alternative conformation was sampled with lower quality from deep MSAs (512:5120), 100 additional structures were sampled each. Finally, as reported in Results, six other proteins were also sampled using the multimer model (alphafold_multimer_v3 weights), 100 conformations each = 600 additional structures, for a total of 34,100 structures. Sampling depths used to produce alternative conformations of fold switchers are reported in Supplementary Table 1. Sampling and success rates for SPEACH-AF, AF-cluster, and AF2 were taken from the supplementary data reported in[19].

### Multimer predictions
CF was run with the same MSA sampling depths as CF-random using the alphafold2_multimer_v3 weights. The number of chains in predicted homo-oligomers matched those from the biological assemblies in the PDB. For diversity, the AF3 server was sampled with up to 5 seeds (1-5) for each of the oligomeric targets. If the alternative conformation could not be found with those seeds, we reported that AF3 did not predict it.

### Other conformational changes
The 14/23 proteins from the SPEACH-AF/OC23 datasets are reported in Supplementary Table 2 and 3. For sampling comparisons, it should be noted that the sampling numbers reported in all cases did not comprise the total number of structures sampled by SPEACH-AF. Rather,

the number of good quality structures produced by SPEACH-AF (by MolProbity score[55]) was reported because the total number of structures generated by SPEACH-AF was not found. MSAs for SPEACH-AF targets were generated using the sequences in their GitHub repository. MSAs for AFsample2 targets were taken directly from their Zenodo repository.

## Coevolutionary analysis

With default settings, MSA Transformer[56] was used to calculate the evolutionary couplings of each MSA uniformly sampled by CF-random at each sampling depth and each recycling step used to produce the best structure corresponding to each alternative conformation. Sequences from –max-seq and –max-extra-seq were combined to make these MSAs. MSA Transformer contacts were superimposed on residue-residue contacts calculated from AlphaFold2 models of a representative dominant and alternative conformation, except 5 that were better produced by the multimer model and 3l5n/2a73 because their large sizes required too much memory to complete the analysis. Structure-based contacts were calculated by finding pairs of heavy atoms from two different residues within 8 Å of each other. Contact intensities for each protein were normalized by the total number (common+dominant+alternative) for each protein for even comparison.

## Sequence association

Protein BLAST[57] was used to search the sequence of Sa1 V90T (PDB ID: 8E6Y) against the PDB. Three hits corresponding to the 3-helix-bundle were identified with PDB IDs: 2FS1, 2MH8, 1GJS. Since these 3 sequences were shorter than 8E6Y, pairwise alignments were constructed by adding gaps to the BLAST alignments at both termini as needed. Each MSA was run through ColabFold1.5.5 with the Sa1 V90T sequence as target, 3 recycles for each MSA. To construct the mixed MSA, an MSA for 1GJS was first constructed using ColabFold1.5.5. This MSA was reregistered with gaps at either or both termini to match the pairwise alignment generated previously. Then, the top 3 sequences from the ColabFold1.5.5-generated MSA for 8E6Y were added to the registered 1GJS MSA. Their UniProt IDs were: A0A7C7DMW5, A0A7C6PUQ5, A0A354ECL0. In all cases, CF was run with all 5 models, 4 random seeds/model.

## Blind mode predictions

For blind mode of CF-random, only an MSA file is required as input. After CF-random has generated all structures from deep and shallow MSAs, a Foldseek[33] database is generated from the predicted ensemble. This database is used to calculate an all against all similarity matrix for the ensemble using the Foldseek structural bitscore. This bitscore is a combination of the Smith–Waterman sequence alignment algorithm, local distance difference test (LDDT), and template modeling (TM)-score. Our approach leverages the combination of LDDT and TM-score within Foldseek's structural bitscore to discern both subtle and large conformational differences between predicted structures because all predicted structures have the same amino acid sequence, discounting the Smith-Waterman algorithm's contribution because the sequences of all structures are identical. Unfolded structures can be produced by the CF-random pipeline–particularly at shallow depths. These structures are filtered by calculating the ensemble's distribution of DSSP assignments and removing outliers from the distribution. The similarity matrix is then reduced using Principal Component Analysis (PCA) to create a lower dimensional space where the distance between points represents how similar/dissimilar the predicted structures are to each other. The HDBSCAN algorithm then clusters similar points together (i.e. points that are both near to each other and of similar density) forming groups of predictions and finally the

K-medoids algorithm is used on each group to select three representative structures. The final result is a subset of the full CF-random ensemble that represents the conformational variance of the predicted structures.

## Colab notebook prediction with blind mode

Blind mode of CF-random is implemented in a Colab notebook. With job name and full-length MSA file, the user can run blind mode without installing on a local machine. Due to the limited resources of free Colab accounts, a set of shallow MSAs is limited as max-seq = 1, 2, 4, and 8, and max-extra-seq = 2 * max-seq. For running ColabFold with the shallow random MSAs, the user can run ColabFold with selected shallow random MSAs. Running the Colab notebook on proteins >300 residues is not recommended with a free Google account.

## E. coli proteome predictions

We ran CF-random on 2126 proteins, representing 65% of proteins ≤300 residues. The remaining 1111 were excluded because we could not generate an MSA deep enough for strong coevolutionary inference by ACE. These proteins were taken from the Refseq[58] E. coli genome, consisting of 5386 proteins. Final predictions in Fig. 7 were refined by running CF with 12–24 recycles instead of 3, resulting in higher pIDDT scores.

## Rosetta energy scores

To estimate the folding energies of E. coli predictions compared to known fold-switching proteins, we calculated the Rosetta Energy Scores (RES) of experimentally characterized fold-switching proteins (Supplementary Table 1) and blind search on the E. coli genome (Supplementary Table 4) using Rosetta[38,39]. The RES of the crystal structures of fold-switching proteins and the predictions of the E.coli genome were calculated after relaxation with the *relax* function[59]. The folding energy difference was obtained following the equation:

$$\Delta E_{folding\ energy} = E_{dominant} - E_{alternative} \qquad (1)$$

## Other software

All protein illustrations were generated with PyMOL[60]. Plots were made with matplotlib[61] and Seaborn[62]. Clustering and PCA were performed with scikit-learn[63]. The scripts for all data analysis were written in Python 3.10 (https://www.python.org/) with following modules: Biopython 1.79 (https://biopython.org/), NumPy 1.23.5 (https://numpy.org/doc/2.1/index.html), tmtools 0.2.0 (https://pypi.org/project/tmtools/), and pandas 2.2.3 (https://pandas.pydata.org/).

## Reporting summary

Further information on research design is available in the Nature Portfolio Reporting Summary linked to this article.

# Data availability

Data generated for the analysis and example data to run scripts and to create figures were deposited on GitHub: https://github.com/ncbi/CF-random_software. The supporting data generated in this study are provided in the Supplementary Information and the Source Data file. The structural data used in this study were taken from the Protein Data Bank are listed below with their accession codes –NMR structures of human XCL1: 1J90 [https://www.rcsb.org/structure/1J9O] and 2JP1; crystal structures of human TRAP1N: 5F5R and 5F3K; crystal structures of E. coli RepE: 1REP and 2Z9O; crystal structure of FraC with lipids 4TSY: [https://www.rcsb.org/structure/4TSY]; crystal structure of the cell cycle regulatory protein Cks1 1QB3; NMR structure of amyloid-beta fibrils: the Osaka mutations 2MVX; NMR structure of pore-forming amyloid-beta tetramers 6RHY; crystal structure of Thermotoga maritima IMPase TM1415 2P3V, chains A and D; NMR structure of Sa1_V90T

8E6Y; NMR structure of the Albumin binding domain of *Streptococcal* Protein G 1GJS; GA-79-MBP CS-rosetta structures 2MH8, and solution NMR structure of PSD-1 2FS1. Unless otherwise stated, all data supporting the results of this study can be found in the article, supplement, and source data files. Source data are provided with this paper.

## Code availability

Code can be found at https://github.com/ncbi/CF-random_software and https://doi.org/10.5281/zenodo.15596182.

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

## Acknowledgements

L.L.P. thanks Carolyn Ott, Marius Clore, and John Jumper for helpful discussions. This work utilized resources from the NIH HPC Biowulf cluster (http://hpc.nih.gov), and it was supported by the Intramural Research Program of the National Library of Medicine, National Institutes of Health (LM202011, L.L.P.).

## Author contributions

Conceptualization: L.L.P., Methodology: M.L., L.L.P., and J.W.S.; Software: M.L, J.W.S., L.L.P., and J.P., Investigation: M.L., J.W.S., J.P., D.C., M.F.C., and L.L.P.,; Data Curation: M.L., J.W.S., D.C., and L.L.P.; Visualization: M.L., J.W.S., D.C., M.F.C., and L.L.P.; Writing – original draft: M.L., L.L.P.; Writing – review & editing: M.L., L.L.P.; Supervision: L.L.P.; Project administration: L.L.P.; Funding acquisition: L.L.P.

## Funding

## Competing interests

The authors declare no competing interests.
