## [Transparent Peer Review file · Nature Communications]

Large-scale predictions of alternative protein conformations by AlphaFold2-based sequence association

Corresponding Author: Dr Lauren Porter

Version 0:

Reviewer comments:

Reviewer #1

(Remarks to the Author)

This is a well written report of a new computational method by which to predict the multiple folded structures of fold switching proteins. It improves upon the predictive power of alpha fold 2, which often only predicts one of the structures adopted by a fold switching protein, while missing the other. Below are a few notes, as they come up during the course of reading the manuscript:

1. Sampling multiple different depths of MSAs and selecting the optimal depth is a clever approach. The authors state that "Typically, sampling occurs at seven depths ranging from 3 to 192 sequences." How was the number seven selected? What would happen if you sampled more, or fewer, depths?

2. The authors write, "We tested CF-random on 92 fold-switching proteins likely in AF2's training set and found that it predicted 32 conformations successfully with <20% of the sampling of other AF2-based methods (Figure 2a), which predicted 24 fold switchers altogether."

This sentence is confusing. I think what it means is that CF random predicted 32 alternate conformations successfully – although the successful prediction could be referring to either dominant or alternate conformations. And, I think what it means is that other AF2 based methods sample >80% more total conformations, yet predict only 24 alternate folds – is that correct? If so, perhaps splitting into 2 sentences and rewording for clarity would be useful, as this is an important point.

3. It seems that the authors tried to improve the 32 successful predictions by combining AF2-multimer with CF-random. This begs the question, could they have turned any failures into successes using this combination approach? It seems the they should try using the combination approach for all 92 of the fold switchers that they tested with CF-random.

4. This sentence is confusing: "Thus, many of these predictions seem to be driven by AF2's learning of protein structure contained rather than couplings unique to alternative conformations" -- the word contained feels out of place, perhaps a typo?

5. Similar to point 3, why was blind mode tested on 32 proteins and not the full set of 92 known fold switchers? Is it guaranteed to fail, and in failing, provide no additional information about the failure mode of CF random? If it is guaranteed to fail, then I suppose it would be worth noting in the text that blind mode is therefore able to identify fold-switching in 25% of the proteins in the entire test set (23/92). While this may at first seem disappointing, on further reflection, this is not an insignificant proportion. If, as estimated by Porter and Looger in 2018, there is as much as 4% of the PDB made up of fold switchers, and there are over 228,000 PDB entries in total, then 25% of the possible 4% of fold switchers would still be over 2,200 proteins that CF-random could find using blind mode in its current state. Sure, it's worth considering that there is some redundancy in the 228,000 entries, but nonetheless it is exciting to consider the possibility that there could be thousands of fold switchers waiting to be discovered, and that this method has more than an outside chance of discovering them.

6. Figure 6 caption: typo, ofent should be often.

Overall, this is an interesting study and I do think that it is worthy of publication. It is useful and encouraging to see the broadening of the utility of the method to include predictions of rigid body motions and local conformational changes as well.

I look forward to even further improvements in the success of methods for prediction the structures of fold switching proteins, and the discovery of new fold switching proteins identified by computational predictive methods.

(Remarks on code availability)

Reviewer #2

(Remarks to the Author)

Porter and co-workers present in this manuscript a study on how ultra-shallow MSA, combined with known structural information, can drive AlphaFold2 (AF2) to predict alternative conformations for fold-switching proteins. The method, termed CF-random, can operate in two different modes. In the first (default) mode, the structures of two reference conformations have to be provided, and the algorithm attempts to recover the alternative conformation, which is closer to the reference conformation not predicted as the dominant conformation by AF2 with a full-depth MSA. In the second (blind) mode, predicted conformations with shallow MSA are clustered using the structural similarity (TM-score) to the dominant conformation and similarity (FoldSeek score) with respect to all possible protein structures, allowing the identification of multiple conformations. Furthermore, the authors show that CF-random sometimes functions by sequence association. While the study conveys interesting observations and insights, CF-random in general lacks the novelty and usefulness needed to significantly advance the prediction of alternative conformations, which is indeed a very important and challenging topic. Detailed comments are provided below.

1) CF-random in its default mode does not have any practical utility, as it requires the input of two reference conformations such as one already know every bit of details on the alternative conformations. This mode serves merely as a test to determine whether AF2 can retrieve or recover the provided information. Consequently, it is not a fair comparison to evaluate CF-random against methods like AF-cluster and AFSample2, which do not require alternative conformations as input to generate them.

2) That said, the authors are strongly encouraged to focus on CF-random in its blind mode, which is indeed a prediction tool. The CF-random in the first mode should be considered for providing theoretical insights and upper-limit in the success rate for CF-random in the blind mode. However, the blind mode is only very briefly described in this manuscript, without systematic benchmarking and detailed analysis. The blind mode relies on clustering using Foldseek score and TM-score, both of which are relatively fuzzy scores. How often does the clear two-cluster behavior, as depicted in Figure 6b, occur? My guess is that it is quite rare. A proper, universal methodology for selecting alternative conformations from the 2D Foldseek/TM-score plot should be explicitly stated, rather than relying on the illustrative example in Figure 6c. Maybe a few different ways can be present and the success rate of each way can be provided.

3) Does one always select only one alternative conformation? Or one can generate multiple ones? How will these differ in the success rate?

4) The FoldSeek score should be between 0.0 and 1.0, with 1.0 being exact match and around 0.5 indicating same fold/family. Why in Figure 6, the normalized Foldseek score can be larger than 1.0 (Fig. 6b and 6c)?

5) Back to the first mode, only 32 out of 92 fold-switch proteins are correctly generated, resulting in a relatively low success rate. Analyzing the cases where predictions fail to generate alternative conformations would provide valuable insights. In addition, considering this 35% success rate as a potential upper limit for related methods, such as CF-random-blind, further analysis and refinement are warranted. Extending the evaluation to other datasets for alternative conformation prediction, such as inward-facing and outward-facing conformations of membrane transporters, could also be helpful.

6) One issue with randomly sampling is the large uncertainty. One could have so many different way of obtaining a MSA combination at the required depths. This is a key issue AF-cluster has solved by using clustering to partition MSAs. In CF-random, are all possible combinations tested? If not, could additional successful cases be recovered by exploring more combinations?

7) The sequence association hypothesis is interesting. It reminds me a decent work on ESM2 "Protein language models learn evolutionary statistics of interacting sequence motifs", which explores failures in structure prediction for protein isoforms. However, the analysis in the current manuscript lacks depth and meaningful insights. Additional tests, such as those involving residue masking, should be conducted to further substantiate and expand upon this hypothesis.

(Remarks on code availability)

The code and data have been provided as described.

Reviewer #3

(Remarks to the Author)

The submission "Efficient predictions of alternative protein conformations by AlphaFold2- based sequence association" discusses advancements in alternative fold/ fold switching prediction based on tweaking the AlphaFold2 (AF2) input. Fold switching involves significant structural changes in proteins, which can be critical for their function. Traditional AF2-based methods, including ColabFold, typically generate a single dominant conformation from deep multiple sequence alignments (MSAs), but do not capture possible additional stable protein conformations, such as those contained in functional cycles, well.

To address this challenge, the authors introduce CF-random, a novel pipeline that combines predictions from both deep and shallow MSA sampling. CF-random is designed to sample very shallow random input MSAs (as few as 3). This approach allows the AF2 network to predict structures based on sparse sequence information, which would be insufficient for any robust coevolutionary inference.

The submitted findings suggest that CF-random provides a more efficient means of sampling alternative conformations, thereby enhancing the predictive capabilities of AI-based protein structure prediction methods. This advancement has significant implications for understanding protein function and the mechanisms underlying fold switching, which are crucial for biomolecular processes or even function.

I find the current submission overall timely and important. The submission itself is very well written with a clear line of thought, a focus on sufficient detail of the technical aspects, and a thorough discussion of all results. While it goes beyond the state of the art, my main concerns deals with the submissions novelty considering the recent advent of many publications covering such conformational heterogeneity: Investigating multiple conformations and functional cycles has been a staple of both the recent and older literature (many MD simulations e.g. on Adenylate Kinase from the 2000s; Martin Weigt and co-workers on Histidine Kinases with DCA/ co-evolution around 2010). Here, in particular the older literature seems poorly covered.

Currently, manipulations of the MSA input of AF2 towards enriching alternative states has been done quite a bit recently, both by the main PI and others. So what are the current submissions main points? The authors list 3 in the current submission: "(1) testing CF-random on 92 fold- switching proteins and 37 other proteins that undergo local conformational fluctuations and rigid body motions, (2) showing that CF-random often works by sequence association, and (3) providing an automatic implementation". All these points seem to be more on the technical side and, while by themselves clearly quite interesting, they do not carry any truly novel biological insight. As an example, while there is discussion of blind predictions I would expect something akin a larger testable dataset of proteins with blindly predicted novel conformations. A bold statement like "for these 1000 protein we believe there might be alternative conformation which is currently unknown". Would that be do-able?

In its current form, I would thus recommend a more technical journal appearing the be more suitable.

Main Concerns:

- no novel biological insight such as blind predictions of unknown folds at scale and how reliable the predictions are likely/ what is the confidence? Maybe not doable "This is a second notable limitation of CF- random: it may predict alternative conformations of proteins erroneously" "it is a weak predictor (35% success rate)" but if you cannot perform blind predictions with CF-random what novel insight is gained?
- poor coverage of approaches before the advent of deep learning
- No comparison or discussion to approaches based on LLM

(Remarks on code availability)

Version 1:

Reviewer comments:

Reviewer #1

(Remarks to the Author)

The revisions have strengthened the manuscript, especially the addition of the predictions of putative fold switching proteins in *E. coli*. Of course, the predictions beg the question of whether any of these proteins would be shown to switch folds experimentally (e.g., by NMR, as the authors suggest), although this may be outside of the scope of this manuscript. In absence of experimental data, it would be useful to use an orthogonal approach to predict whether these predicted fold switching structures would actually fold. The authors have used alpha fold's pLDDT as a numeric metric, but it would be interesting and compelling to have an additional metric or two that are calculated in different ways. For example, one could predict delta G of folding for each structure in the fold switched pairs. Then, one could assess whether predicted folding energies indicate that each fold would form stably enough to be experimentally detected, and whether folding energies are similar to one another (suggesting the possibility of fold switching).

(Remarks on code availability)

Reviewer #2

(Remarks to the Author)

The manuscript has been significantly improved, particularly with the updated Figure 6 and the newly added Figure 7. The authors have done a much better job of explaining both the CF-random and CF-random-blind methods. I did notice that some technical details related to CF-random-blind have changed between the previous and current versions, which I assume reflects the authors' optimization of the method to achieve better performance. The additional screening of the E. coli proteome is also both interesting and potentially useful.

I agree with other reviewers that the authors still do not sufficiently discuss their results in the context of related work in the field. This issue persists, but given that this is a rapidly evolving field, it may not be possible to mention every related method. Overall, I recommend the publication of this manuscript in its current form. It is a timely submission and will make a valuable contribution to this exciting field.

(Remarks on code availability)

Reviewer #3

(Remarks to the Author)

The authors have substantially addressed my earlier concerns. As my main concern was that of sufficient biological novelty, I particularly appreciate the expansion of the scope of their study to include a large-scale blind prediction using their improved CF-random "Blind Mode". I applaud the authors' tenacity to expand the scope of the current submission. Shortly, in this revision, they have applied their method to over 2000 E. coli proteins and identified more than 50 putative fold switchers. This blind prediction experiment represents a meaningful step toward demonstrating that CF-random can generate novel, testable/ verifiable biological insights rather than just being merely a technical advancement.

Regarding the discussion of previous approaches, the authors include now literature the literature on other methods—such as molecular dynamics (MD) simulations. I can relate to their argument that it was not extensively reviewed previously because these methods do not scale to the level required for genome-wide fold-switching predictions. They note that while MD simulations have been successful in modeling alternative conformations, their high computational cost (especially for fold switching events that occur on the scale of seconds) makes them impractical for large-scale screening. This rationale is now clearly articulated in the revised Introduction, along with a discussion on why LLM-based methods like ESMFold, which have been tested in this context, are less robust than AF2 for predicting alternative conformations.

Overall, the authors have therefore effectively addressed my concerns by demonstrating that CF-random is capable of generating novel predictions at a scale that traditional methods cannot achieve. The added discussion on the limitations of MD simulations and ESMFold helps to justify the focus on their AF2- based predictions, particularly in the context of genome-wide analyses. I believe the revised manuscript is now well-positioned for publication.

ps:

As a remaining very minor point that could be addressed even during a possible proof-stage I would call MD and similar methods physics-based not "physically based" (p.1).

(Remarks on code availability)

Version 2:

Reviewer comments:

Reviewer #1

(Remarks to the Author)

The revisions have strengthened the manuscript. Recommend publication.

(Remarks on code availability)

Reviewer #1 (Remarks to the Author)

This is a well written report of a new computational method by which to predict the multiple folded structures of fold switching proteins. It improves upon the predictive power of alpha fold 2, which often only predicts one of the structures adopted by a fold switching protein, while missing the other. Below are a few notes, as they come up during the course of reading the manuscript:

We thank the Reviewer for their positive comments and good questions.

1. Sampling multiple different depths of MSAs and selecting the optimal depth is a clever approach. The authors state that “Typically, sampling occurs at seven depths ranging from 3 to 192 sequences.” How was the number seven selected? What would happen if you sampled more, or fewer, depths?

This work seeks to balance success rate with efficient sampling so that large-scale predictions of alternative conformations can be achieved. Previous work by del Alamo and Mchaourab (eLife 2022) suggested sampling alternative conformations with `--max-seq=2n`, `--max-extra-seq=2n+1`, which was our starting point. During this project, we tried other ratios, especially at shallow depths (e.g. `--max-seq=n`, `--max-extra-seq = n`) to see if other alternative conformations could be recovered. They weren't. Furthermore, sampling deeper MSAs is more prone to returning the dominant conformation (see Figure 2b—fewer alternative conformations are recovered with deeper MSAs). Thus, we found these 7 depths to be optimal for achieving the aim of this algorithm.

2. The authors write, “We tested CF-random on 92 fold-switching proteins likely in AF2's training set and found that it predicted 32 conformations successfully with <20% of the sampling of other AF2-based methods (Figure 2a), which predicted 24 fold switchers altogether.”

This sentence is confusing. I think what it means is that CF random predicted 32 alternate conformations successfully – although the successful prediction could be referring to either dominant or alternate conformations. And, I think what it means is that other AF2 based methods sample >80% more total conformations, yet predict only 24 alternate folds – is that correct? If so, perhaps splitting into 2 sentences and rewording for clarity would be useful, as this is an important point.

We thank the Reviewer for this suggestion and changed that part as follows:

We tested CF-random on 92 fold-switching proteins likely in AF2's training set and found that it predicted both the dominant and alternative conformations of 32 fold switchers successfully. Further, CF-random sampled >80% fewer structures compared to other AF2-based methods (Figure 2a), which predicted 24 fold switchers altogether.

3. It seems that the authors tried to improve the 32 successful predictions by combining AF2-multimer with CF-random. This begs the question, could they have turned any failures into successes using this combination approach? It seems the they should try using the combination approach for all 92 of the fold switchers that they tested with CF-random.

Running CF-random on all 92 fold switchers using the multimer model added 1 additional hit, which we report in the MS as follows:

After running CF-random with the multimer model on all 92 fold-switching proteins, only one additional alternative conformation was found: the dimeric form of bone morphogenic protein inhibitor DAN from Mus musculus (PDB ID: 4jph). It was not included among the successes since it was not identified by CF-random in monomer mode.

We did not update the numbers in Figure 2a because all other predictive methods used only monomeric models, and the 32 initial CF-random hits were also achieved with the monomer-ptm model. Thus, it seemed like comparing numbers resulting from monomer presets was a fairer comparison.

4. This sentence is confusing: "Thus, many of these predictions seem to be driven by AF2's learning of protein structure contained rather than couplings unique to alternative conformations" -- the word contained feels out of place, perhaps a typo?

We thank the Reviewer for catching this typo. We removed "contained."

5. Similar to point 3, why was blind mode tested on 32 proteins and not the full set of 92 known fold switchers? Is it guaranteed to fail, and in failing, provide no additional information about the failure mode of CF random? If it is guaranteed to fail, then I suppose it would be worth noting in the text that blind mode is therefore able to identify fold-switching in 25% of the proteins in the entire test set (23/92). While this may at first seem disappointing, on further reflection, this is not an insignificant proportion. If, as estimated by Porter and Looger in 2018,

there is as much as 4% of the PDB made up of fold switchers, and there are over 228,000 PDB entries in total, then 25% of the possible 4% of fold switchers would still be over 2,200 proteins that CF-random could find using blind mode in its current state. Sure, it's worth considering that there is some redundancy in the 228,000 entries, but nonetheless it is exciting to consider the possibility that there could be thousands of fold switchers waiting to be discovered, and that this method has more than an outside chance of discovering them.

The first draft of our manuscript was unclear on this issue; here, we clarify. The only difference between blind mode and default mode is the method of identifying alternative conformations from CF-random. We ran blind mode on all 92 fold switchers, and it had an 81% success rate (26/32) and an overall success rate of 28%, which is not reported in the manuscript:

Blind mode successfully identified both conformations of fold switchers in 81% of cases identified in default mode (26/32), corresponding to a 28% success rate among all 92 fold switchers.

Notably, blind mode did not produce any new fold switchers—only the ones we had identified previously. Since the only thing that changes between modes is the randomly selected random seed (both selected at random), this suggests that 5 random seeds/target is enough sampling for CF-random to converge.

To the other point, in the final section of this revised manuscript, we report predictions of fold switchers from a large portion of the *E. coli* proteome. We hope that CF-random will facilitate new discoveries.

6. Figure 6 caption: typo, ofent should be often.
Fixed.

Overall, this is an interesting study and I do think that it is worthy of publication. It is useful and encouraging to see the broadening of the utility of the method to include predictions of rigid body motions and local conformational changes as well. I look forward to even further improvements in the success of methods for prediction the structures of fold switching proteins, and the discovery of new fold switching proteins identified by computational predictive methods.(Remarks on code availability)

Once again, we thank the Reviewer for their positive comments.

Reviewer #2 (Remarks to the Author)

Porter and co-workers present in this manuscript a study on how ultra-shallow MSA, combined with known structural information, can drive AlphaFold2 (AF2) to predict alternative conformations for fold-switching proteins. The method, termed CF-random, can operate in two different modes. In the first (default) mode, the structures of two reference conformations have to be provided, and the algorithm attempts to recover the alternative conformation, which is closer to the reference conformation not predicted as the dominant conformation by AF2 with a full-depth MSA. In the second (blind) mode, predicted conformations with shallow MSA are clustered using the structural similarity (TM-score) to the dominant conformation and similarity (FoldSeek score) with respect to all possible protein structures, allowing the identification of multiple conformations. Furthermore, the authors show that CF-random sometimes functions by sequence association. While the study conveys interesting observations and insights, CF-random in general lacks the novelty and usefulness needed to significantly advance the prediction of alternative conformations, which is indeed a very important and challenging topic. Detailed comments are provided below.

We thank the Reviewer for their comments, which allow us to explain better why we think CF-random is a significant advance.

Before addressing specific comments below, it is important to highlight that CF-random requires much less sampling than other methods. When using a method to query thousands of genomic sequences, as we do in this revision, less sampling enhances performance and allows for easier analysis of output structures. To our knowledge, this is the first paper to blindly predict numerous three-dimensional models of fold-switching proteins from an organism's proteome, to present detailed curation methods, and to suggest a rigorous estimate for the frequency of fold switching in a genome. Thus, in our view this is an exciting and significant contribution

1) CF-random in its default mode does not have any practical utility, as it requires the input of two reference conformations such as one already know every bit of details on the alternative conformations. This mode serves merely as a test to determine whether AF2 can retrieve or recover the provided information. Consequently, it is not a fair comparison to evaluate CF-random against methods

like AF-cluster and AFSample2, which do not require alternative conformations as input to generate them.

CF-random successfully predicts 32 alternative conformations with no reference inputs. We have remade Figure 1 and amended the text to clarify this point. Specifically, the reference conformations are not needed to steer AF2 to produce an alternative conformations. However, all 32 alternative conformations we report are generated using 5 random seeds at 5 sampling depths (200 predictions in all) with no input information required. The authors of AFSample2 tuned their MSA randomization for this purpose using PDBs as references (Figure 1 in <https://www.biorxiv.org/content/10.1101/2024.05.28.596195v1.full.pdf>). Further, both AF-cluster (Figures 1-3 and some supplementary) and AFSample2 (Figures 1 and 3) also used reference PDBs to evaluate their predictions.

AF-cluster also compares ensembles to reference PDBs and provides code to do so (see the GitHub site). Although some figures in the manuscript show “PCA on Contacts” on the axes, the code was not provided, and the answers for the alternative conformations were already known. Thus, it is likely these graphs were protein-specific and may not generalize. AFSample2 does provide a blind search method to select alternative conformations of open/closed conformations, however to the best of our knowledge, that was not used to assess its overall performance.

With this in mind, CF-random outperforms AF-cluster and AFsample2 in both its success rate and efficiency. AF-cluster’s success rate in our Figure 2 was determined using the same methodology: comparing its output to the same experimentally determined structures used to benchmark CF-random (see Figure 1 and related text in <https://www.nature.com/articles/s41467-024-51801-z>).

2) That said, the authors are strongly encouraged to focus on CF-random in its blind mode, which is indeed a prediction tool. The CF-random in the first mode should be considered for providing theoretical insights and upper-limit in the success rate for CF-random in the blind mode. However, the blind mode is only very briefly described in this manuscript, without systematic benchmarking and detailed analysis. The blind mode relies on clustering using Foldseek score and TM-score, both of which are relatively fuzzy scores. How often does the clear two-cluster behavior, as depicted in Figure 6b, occur? My guess is that it is quite rare. A proper, universal methodology for selecting alternative conformations from the 2D Foldseek/TM-score plot should be explicitly stated, rather than relying on the

illustrative example in Figure 6c. Maybe a few different ways can be present and the success rate of each way can be provided.

We are glad that the Reviewer appreciates blind mode. Aligned with their suggestions, we have improved it and used it to predict new fold-switching proteins from thousands of *E. coli* proteins. Blind mode now selects representative conformations and stores them in a pymol (.pse) file for easy comparison. Thus, no user input is required to select representative conformations. This blind mode's success rate was higher than our previous blind mode (81% vs 72%). We designed a new blind mode because we tested other methods (PCA on contacts and TM-score vs. pLDDT) and found them to be insensitive to local conformational changes, which many fold switchers undergo.

Further, CF-random's ability to predict plausible new fold switchers attests to the utility and sensitivity of our Blind Mode. For comparison, AF-cluster predicted 1 putative alternative fold out of 628 (0.2% hit rate). CF-random's predicted 52/2126 (2.8% hit rate).

3) Does one always select only one alternative conformation? Or one can generate multiple ones? How will these differ in the success rate?

Blind mode selects more than one predicted alternative conformation, and it is up to the user to decide which they find plausible. In this Revision, we include the quality checks we used to ensure predictions were as reliable as possible.

4) The FoldSeek score should be between 0.0 and 1.0, with 1.0 being exact match and around 0.5 indicating same fold/family. Why in Figure 6, the normalized Foldseek score can be larger than 1.0 (Fig. 6b and 6c)?

We updated Figure 6 with a new Blind Mode, so this is no longer an issue.

5) Back to the first mode, only 32 out of 92 fold-switch proteins are correctly generated, resulting in a relatively low success rate. Analyzing the cases where predictions fail to generate alternative conformations would provide valuable insights. In addition, considering this 35% success rate as a potential upper limit for related methods, such as CF-random-blind, further analysis and refinement are warranted. Extending the evaluation to other datasets for alternative conformation prediction, such as inward-facing and outward-facing conformations of membrane transporters, could also be helpful.

We attempted that in previous work on fold switchers (Chakravarty, et al. Nat. Comms. 2024) and did not find any obvious reasons why it did not work. Other groups who have worked on transporters have not reported reasons why either (e.g. AFSample2 and SPEACH-AF). We suspect that the failures represent blind spots in AF2's learning of protein structure, which is consistent with its recent failures to predict new folds, which we describe here: <https://www.sciencedirect.com/science/article/pii/S0959440X24002008>.

6) One issue with randomly sampling is the large uncertainty. One could have so many different way of obtaining a MSA combination at the required depths. This is a key issue AF-cluster has solved by using clustering to partition MSAs. In CF-random, are all possible combinations tested? If not, could additional successful cases be recovered by exploring more combinations?

CF-random samples alternative conformations more successfully than AF-cluster (Figure 2) and with higher efficiency. Further, we have shown in previous work that AF-cluster worsens predictions of alternative conformations compared to random sampling (<https://www.nature.com/articles/s41586-024-08267-2>). Thus, we do not think AF-cluster is a good solution to the problem.

While more sampling with CF-random may sometimes yield more hits, this suffers from diminishing returns. For fold switchers, sampling twice as much did not yield more hits. This is consistent with what Bryant and Noé observed with their implementation of AlphaFold2 (CFold): extensive sampling generally did not improve its success rate (<https://www.nature.com/articles/s41467-024-51507-2>).

7) The sequence association hypothesis is interesting. It reminds me a decent work on ESM2 "Protein language models learn evolutionary statistics of interacting sequence motifs", which explores failures in structure prediction for protein isoforms. However, the analysis in the current manuscript lacks depth and meaningful insights. Additional tests, such as those involving residue masking, should be conducted to further substantiate and expand upon this hypothesis.

Although both papers seek to interpret what is going on in neural networks, they are not the same. ESM2 is an unsupervised network trained on protein sequences, making it much easier to interpret. AlphaFold2 is more complex because it is

supervised and therefore depends on structures in the training set as well as sequence data. Our previous work supports the sequence association hypothesis by showing that AF2 has memorized structures during training and associates them with similar sequences (<https://www.nature.com/articles/s41467-024-51801-z>). While we agree that a more in-depth study is warranted, that is outside the scope of this manuscript. We developed CF-random to predict alternative conformations from proteomes, which we show here for the first time.

In another study that we recently submitted, we found that for 3 fold switchers (XCL1, RfaH, and KaiB), AF2 associates alternative conformations with sparse sequence patterns of 2-5 amino acids. This is consistent with the hypothesis we propose.

Further, we are working on a larger follow-up study on AF2-based sequence association of proteins in general, not just fold switchers. So far, we have tested over 30 single-folding proteins with different architectures and lengths and found that, 88% of them are predicted accurately and confidently with 2-sequence MSAs (target + 1 homologous sequence). We are currently extending this study to >1000 diverse targets, but our preliminary results suggest that AF2 does indeed frequently work by sequence association.

(Remarks on code availability)

The code and data have been provided as described.

Reviewer #3 (Remarks to the Author):

The submission “Efficient predictions of alternative protein conformations by AlphaFold2- based sequence association“ discusses advancements in alternative fold/ fold switching prediction based on tweaking the AlphaFold2 (AF2) input. Fold switching involves significant structural changes in proteins, which can be critical for their function. Traditional AF2-based methods, including ColabFold, typically generate a single dominant conformation from deep multiple sequence alignments (MSAs), but do not capture possible additional stable protein conformations, such as those contained in functional cycles, well.

To address this challenge, the authors introduce CF-random, a novel pipeline that combines predictions from both deep and shallow MSA sampling. CF-random is designed to sample very shallow random input MSAs (as few as 3).

This approach allows the AF2 network to predict structures based on sparse sequence information, which would be insufficient for any robust coevolutionary inference.

The submitted findings suggest that CF-random provides a more efficient means of sampling alternative conformations, thereby enhancing the predictive capabilities of AI-based protein structure prediction methods. This advancement has significant implications for understanding protein function and the mechanisms underlying fold switching, which are crucial for biomolecular processes or even function.

I find the current submission overall timely and important. The submission itself is very well written with a clear line of thought, a focus on sufficient detail of the technical aspects, and a thorough discussion of all results. While it goes beyond the state of the art, my main concerns deals with the submissions novelty considering the recent advent of many publications covering such conformational heterogeneity: Investigating multiple conformations and functional cycles has been a staple of both the recent and older literature (many MD simulations e.g. on Adenylate Kinase from the 2000s; Martin Weigt and co-workers on Histidine Kinases with DCA/ co-evolution around 2010). Here, in particular the older literature seems poorly covered.

Currently, manipulations of the MSA input of AF2 towards enriching alternative states has been done quite a bit recently, both by the main PI and others. So what are the current submissions main points? The authors list 3 in the current submission: “(1) testing CF-random on 92 fold- switching proteins and 37 other proteins that undergo local conformational fluctuations and rigid body motions, (2) showing that CF-random often works by sequence association, and (3) providing an automatic implementation”. All these points seem to be more on the technical side and, while by themselves clearly quite interesting, they do not carry any truly novel biological insight. As an example, while there is discussion of blind predictions I would expect something akin a larger testable dataset of proteins with blindly predicted novel conformations. A bold statement like “for these 1000 protein we believe there might be alternative conformation which is currently unknown”. Would that be do-able?

We thank the Reviewer for their kind words and thoughtful suggestions, which we address under their Main Concerns.

In its current form, I would thus recommend a more technical journal appearing to be more suitable.

Main Concerns:

-no novel biological insight such as blind predictions of unknown folds at scale and how reliable the predictions are likely/ what is the confidence? Maybe not doable “This is a second notable limitation of CF- random: it may predict alternative conformations of proteins erroneously“ “it is a weak predictor (35% success rate)“ but if you cannot perform blind predictions with CF-random what novel insight is gained?

We thank the Reviewer for this suggestion. We have now improved Blind Mode and used it to predict >50 putative fold-switching proteins out of >2000 *E. coli* proteins, reported in 2 new sections in Results. We carefully curated this list based on our knowledge of fold switching proteins and believe that at least some of the hits may prove correct. To our knowledge, this is the first paper to generate and blindly select dozens of putative fold switchers from among thousands. This attests to the utility and sensitivity of our Blind Mode. For comparison, AF-cluster found 1 putative alternative fold out of 628 (0.2% hit rate). CF-random’s rate was 2.8%.

-poor coverage of approaches before the advent of deep learning

The reason we did not discuss previous approaches is because they do not scale nearly as well as AI-based methods. Our object in designing CF-random was to run it on thousands of proteins, which we do in this revision. To our knowledge, MD simulations could not be used to predict new fold switchers at this scale because they require too many computational resources. This is especially true since fold switching occurs on a timescale of seconds. We explain this in our revised Introduction:

Alternative protein conformations can play critical roles in protein function and regulation¹⁻³. These alternative conformations can be accessed by rigid body reorientations, local fluctuations, or remodeling secondary and/or tertiary structure (fold switching)⁴. Physically-based methods, such as molecular dynamics (MD) simulations, have successfully modeled alternative conformations⁵⁻⁸, but they require too much computational power to predict conformational changes on a large scale. Furthermore, some conformational changes, such as fold switching, occur on a timescale of seconds⁹⁻¹¹, prohibitively long for MD to reasonably access alternative conformations if they were not known previously.

Coevolutionary analysis can also be used to predict alternative conformations, but it does not produce 3D models (though it could be combined with a method like AWSEM, which also does not scale as well as deep learning models). For these reasons, our manuscript focuses on DL models more than earlier work.

-No comparison or discussion to approaches based on LLM

In the 3 years we have spent developing the tools needed to predict fold switchers from genomes, we tried many methods, including LLMs. It became apparent early on that ESMFold is less robust than AF2 for predicting alternative conformations of fold switchers. In fact, extensively masking the fold-switching sequences of some known fold switchers, such as RfaH, yielded correct predictions of alternative conformations, indicating structure memorization. This limits ESMFold's utility in predicting alternative conformations. Others have also found that ESMFold has limited utility in predicting alternative conformations of proteins including fold switchers: <https://www.biorxiv.org/content/10.1101/2023.12.16.571997v1.full> and that it has memorized structures <https://www.pnas.org/doi/abs/10.1073/pnas.2406285121>. That is why we did not discuss LLMs here.

Reviewer #1 (Remarks to the Author):

The revisions have strengthened the manuscript, especially the addition of the predictions of putative fold switching proteins in *E. coli*. Of course, the predictions beg the question of whether any of these proteins would be shown to switch folds experimentally (e.g., by NMR, as the authors suggest), although this may be outside of the scope of this manuscript. In absence of experimental data, it would be useful to use an orthogonal approach to predict whether these predicted fold switching structures would actually fold. The authors have used alpha fold's pLDDT as a numeric metric, but it would be interesting and compelling to have an additional metric or two that are calculated in different ways. For example, one could predict ΔG of folding for each structure in the fold switched pairs. Then, one could assess whether predicted folding energies indicate that each fold would form stably enough to be experimentally detected, and whether folding energies are similar to one another (suggesting the possibility of fold switching).

We appreciate the Reviewer's suggestion and agree that experimental testing is the logical next step for these predictions. Though experiments are outside the scope of this work, we intend to test all targets. We are optimistic that some predictions will be correct, and we hope to have interesting results to report soon.

In the meantime, it seems important to point out that experimental support for some of these 52 predictions exists already. In the last two paragraphs on Page 13, we say Tier 1 predictions were "supported by experiment or had a strong biological basis for the conformational change." These are indicated in Table S4. Two of them are RfaH and MinE, experimentally confirmed fold switchers in our list of 92, which blind mode picked up with no prior knowledge. Another is a fimbrial protein homologous to another fold switcher in our dataset. A fourth is a Cro protein from a family where fold switching has been observed to occur over the course of evolution (PMID: 30051937). These hits assure us that CF-random is giving some reasonable leads.

Serious energy calculations—done properly and accurately—would require work beyond the scope of this paper. In recent work, we showed that AlphaFold confidently predicts incorrect conformations of some proteins (PMID: 39972235). We have run 5 microsecond MD simulations on these incorrect models and see no evidence of unfolding at various temperatures. Thus, more complex (and computationally intensive) approaches would be needed to identify

factors that distinguish confident-but-incorrect AF2 predictions from correct ones. Such an analysis would merit its own manuscript.

As a quick-and-dirty estimate, we scored each pair of 52 models with the Rosetta scoring function and calculated their relative energy differences; a similar approach was applied recently to evaluate AF2 models (<https://www.biorxiv.org/content/10.1101/2023.09.05.556364v2.abstract>). The distributions of relative energies for these models were well within the range expected for experimentally determined fold switchers (added as Figure S8). We interpret this to mean that both models of each fold-switched pair are plausible and mention that in the manuscript in the last paragraph on page 13:

Supporting the plausibility of these predictions, the estimated relative energies between each pair of putative fold switchers were in the same range as fold switchers determined by experiment (Figure S8).

Reviewer #2 (Remarks to the Author):

The manuscript has been significantly improved, particularly with the updated Figure 6 and the newly added Figure 7. The authors have done a much better job of explaining both the CF-random and CF-random-blind methods. I did notice that some technical details related to CF-random-blind have changed between the previous and current versions, which I assume reflects the authors' optimization of the method to achieve better performance. The additional screening of the E. coli proteome is also both interesting and potentially useful. We thank the Reviewer for their positive comments. Indeed, we optimized blind mode during revision.

I agree with other reviewers that the authors still do not sufficiently discuss their results in the context of related work in the field. This issue persists, but given that this is a rapidly evolving field, it may not be possible to mention every related method. Overall, I recommend the publication of this manuscript in its current form. It is a timely submission and will make a valuable contribution to this exciting field.

This is indeed a rapidly evolving field, making it difficult to mention every related method. Nevertheless, to cover the field more widely, we have now discussed several more methods in the Discussion:

Some recently developed methods^{45,46} generate alternative conformations using new techniques such as the Distributional Graphoformer⁴⁷ and flow matching⁴⁷, though these have not been tested on many fold-switching proteins. A diffusion-based model called EigenFold was tested on fold switchers, but its performance was weak⁴⁸. Several previous methods have relied on coevolutionary information to predict alternative conformations, for both AlphaFold2¹⁷ and ESMFold⁴⁹.

Reviewer #3 (Remarks to the Author):

The authors have substantially addressed my earlier concerns. As my main concern was that of sufficient biological novelty, I particularly appreciate the expansion of the scope of their study to include a large-scale blind prediction using their improved CF-random “Blind Mode”. I applaud the authors’ tenacity to expand the scope of the current submission. Shortly, in this revision, they have applied their method to over 2000 E. coli proteins and identified more than 50 putative fold switchers. This blind prediction experiment represents a meaningful step toward demonstrating that CF-random can generate novel, testable/verifiable biological insights rather than just being merely a technical advancement.

Regarding the discussion of previous approaches, the authors include now literature the literature on other methods—such as molecular dynamics (MD) simulations. I can relate to their argument that it was not extensively reviewed previously because these methods do not scale to the level required for genome-wide fold-switching predictions. They note that while MD simulations have been successful in modeling alternative conformations, their high computational cost (especially for fold switching events that occur on the scale of seconds) makes them impractical for large-scale screening. This rationale is now clearly articulated in the revised Introduction, along with a discussion on why LLM-based methods like ESMFold, which have been tested in this context, are less robust than AF2 for predicting alternative conformations.

Overall, the authors have therefore effectively addressed my concerns by demonstrating that CF-random is capable of generating novel predictions at a scale that traditional methods cannot achieve. The added discussion on the limitations of MD simulations and ESMFold helps to justify the focus on their AF2-based predictions, particularly in the context of genome-wide analyses. I believe the revised manuscript is now well-positioned for publication.

We thank the Reviewer for their positive comments.

ps:

As a remaining very minor point that could be addressed even during a possible proof-stage I would call MD and similar methods physics-based not “physically based” (p.1).

Fixed.